# The study of the impact of polar warming on global atmospheric circulation and mid-latitude baroclinic waves using a laboratory analog

Andrei Sukhanovskii[1,3], Andrei Gavrilov[2], Elena Popova[1], and Andrei Vasiliev[1]

[1]Institute of Continuous Media Mechanics UB RAS, 614013, Ac. Korolev Street,1, Perm, Russia
[2]Institute of Thermophysics SB RAS, 630090, Ac. Lavrentieva ave.1, Novosibirsk, Russia
[3]Perm State University, 614068, Bukireva Street.15, Perm, Russia

**Correspondence:** Andrei Sukhanovskii (san@icmm.ru)

**Abstract.** The results of experimental and numerical modeling of Arctic warming in a laboratory dishpan configuration are presented. The Arctic warming is reproduced by varying the size of a local cooler in the "atmospheric" regime, in which the flow structure is similar to the general atmospheric circulation. The laboratory Arctic warming results in a relatively weak response of the meridional and zonal circulation except in the polar region, where the polar cell analog becomes weaker, shifts closer to the middle radii, and is mainly located in the upper layer. The structure of analogs of Hadley and Ferrel cells is the same for all considered configurations. The decrease in the velocity of the zonal flow (analog of westerly wind) and the change in baroclinic wave activity at laboratory middle latitudes was less than 10%. The most important result of this study is a noticeable transformation of the mean temperature field. Namely, the central region and most of the lower layer become warmer, while most of the upper layer and the peripheral (equatorial) part of the lower layer become colder. The nature of this phenomenon is closely related to the changes in radial heat fluxes. The weakening and upward shift of the polar cell analog caused by laboratory Arctic warming provides a significant reduction in the negative heat flux near the bottom. This inevitably leads to a temperature increase in the bottom layer. It is also shown that Ekman pumping due to non-slip boundary conditions at the surface of the cooler has a strong influence on the structure and intensity of the polar cell analog.

## 1 Introduction

Baroclinic waves define the mid-latitude weather, providing meridional transfer of heat and angular momentum (Schneider, 2006). The formation of mid-latitude baroclinic waves is strongly linked to the instability of the axisymmetric zonal flow produced by the Hadley circulation. The study of baroclinic waves in a full statement is an extremely complex problem due to an essentially non-linear nature of the process, which depends on different factors such as rotation, solar heating, and surface topography. The need to reveal robust, intrinsic features of atmospheric baroclinic waves stimulates several laboratory and numerical studies using simplified models (Read et al., 2014). These studies produced very fruitful results to understand the nature and different characteristics of baroclinic waves and showed that the main factors responsible for the formation of baroclinic waves are rotation, cooling and heating. The strong dependence of baroclinic waves on the meridional temperature

difference raises questions about possible scenarios of their evolution due to changes in the global temperature distribution under the influence of various complex processes, including anthropogenic forcing (Hansen and Stone, 2016). In particular, the remarkable warming amplification over the Arctic pole (Arctic amplification) results in a decrease of the temperature contrast between the pole and the equator (You et al., 2021). Arctic amplification can lead to complex chains of processes that strongly influence large-scale circulation and the likelihood of weather extremes (Overland et al., 2016). The prediction based on numerical calculations suggests that the Arctic amplification will continue (Wallace et al., 2016). The ice-temperature feedback in the Arctic increases the likelihood of further rapid warming and sea ice loss, and may affect atmospheric circulation in the polar region and mid-latitudes (Screen and Simmonds, 2010; Cohen et al., 2014).

There are intense debates about connection between the intensity and meandering (or waviness) of mid-latitude zonal flow and Arctic amplification. According to one hypothesis (Francis and Vavrus, 2012, 2015), a decrease in the meridional temperature gradient due to Arctic amplification weakens the mid-latitude zonal wind and, as a consequence, leads to an increase in the amplitude of mid-latitude waves and waviness of the zonal circulation. The other hypothesis is directly opposite and suggests that there is no significant influence of the Arctic amplification on the waviness of the mid-latitude circulation in observations or models, and the observed transformation of westerly winds is the result of the internal variability of the mid-latitude circulation (Blackport and Screen, 2020). The Polar Amplification Model Intercomparison Project (Smith et al., 2022) and very large-ensemble climate model simulations (Ye et al., 2024) shows that the winter tropospheric circulation response to projected Arctic sea-ice loss is robust but weak compared to interannual variability. This includes equatorward shift of storm-tracks, weakening of mid-latitude westerlies and storm-track activities. The controversial conclusions about the influence of the Arctic amplification are partly based on the use of different data, models, approaches and metrics, so the joint efforts of the scientific community are needed to reach a generally accepted understanding of the problem (Overland et al., 2016; Stuecker et al., 2018).

Laboratory modeling can help to understand the main tendency of baroclinic wave evolution due to the variations of heating and cooling. There are two main alternative laboratory approaches to the study of baroclinic waves, the so-called dishpan configuration (Fultz et al., 1959) and annulus configuration (Hide, 1953). The main differences are related to the geometry of the fluid layer and the realization of heating and cooling. The dishpan configuration is a cylindrical vessel (usually horizontally extended) with the rim heating at the bottom periphery and cooling in the center, while the annulus configuration is a cylindrical gap between inner and outer cylinders with isothermal vertical walls (inner walls is cold, outer wall is hot). The specifics and comparison of the results for both configurations were described and discussed in (Harlander et al., 2023). The main difference is the tendency of the baroclinic waves to show more intrinsic instability for the dishpan configuration.

Recently, a polar warming scenario has been considered in laboratory experiments carried out in the annulus configuration (Rodda et al., 2022). It was shown that a progressive decrease of the meridional temperature difference slows down the eastward propagation of the jet stream and complicates its structure. Temperature variability decreases relative to the laboratory Arctic warming only at locations representing the polar and mid-latitudes of the Earth, which are influenced by the jet stream. In the subtropical region south of the simulated jet, the trend is reversed. The reduced variability leads to narrower temperature distributions and weaker extreme events, but the frequency of such events increases in the polar and mid-latitudes

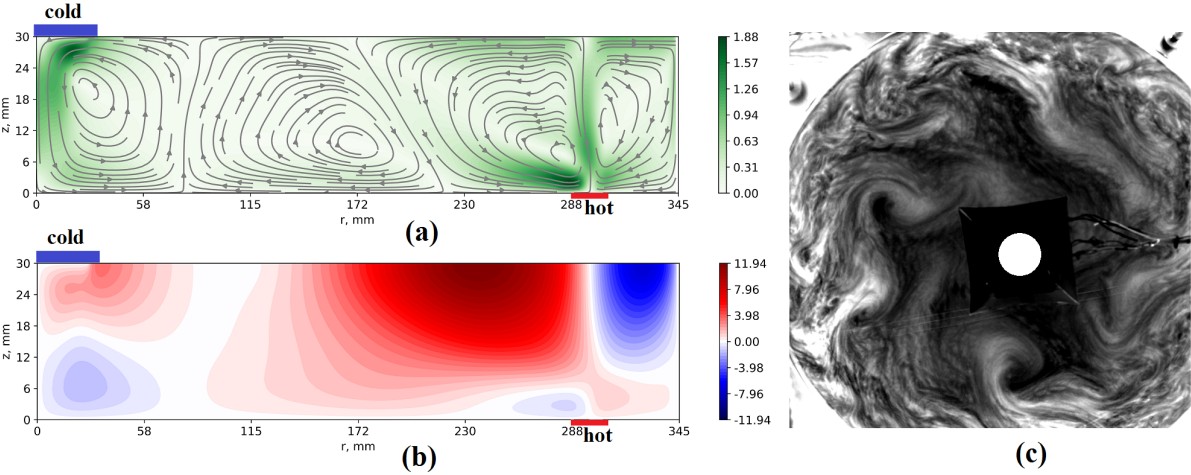

**Figure 1.** The typical structure of the laboratory analog of atmospheric circulation (in the rotating frame). (a) Streamlines of the mean meridional circulation, the absolute values of velocity (in $\mathrm{mm/s}$) are shown by the color shading (b) Distribution of the mean zonal velocity, in $\mathrm{mm/s}$ (numerical simulation), (c) Instantaneous image of the flow structure (experiment, visualization by aluminum powder, view from above). (Colour online)

and decreases towards the subtropics with decreasing meridional temperature difference. The obtained results showed good qualitative agreement with the National Centers for Environmental Prediction (NCEP) reanalysis data.

In the present study we have conducted a series of experiments and numerical simulations for the Arctic warming scenario in the dishpan configuration (Sukhanovskii et al., 2023; Vasiliev et al., 2023). The laboratory model of the general atmospheric circulation is characterized by the three-cell structure (analogs of Hadley, Ferrel and polar cells), the intense zonal flow in the middle radii (analog of westerly winds), and the developed system of baroclinic waves with dominant wave numbers from $m = 4$ to $m = 8$. The combined laboratory and numerical modeling of the Arctic amplification in a relatively simple statement

can provide valuable information about the relationship between the Arctic amplification and the mid-latitude zonal flow and baroclinic waves.

The structure of the paper is as follows. The statement of the problem and governing parameters are given in section 2, the experimental set-up and mathematical model are described in section 3. The main results, including description of the flow structure (subsection 4.1) and heat transfer analysis (subsection 4.2) are presented in section 4. A summary and conclusions

are given in section 5.

## 2    Statement of the problem and governing parameters

In the present study we consider a shallow rotating cylindrical layer of fluid with a localized heater at the bottom in the periphery and a localized cooler in the central part of the upper boundary. The rim heater mimics the equatorial heating and

the disc cooler mimics the north pole cooling. The rim heater is intentionally shifted from the sidewall to reduce the influence of the non-slip vertical boundaries. Boundary conditions of the second type (constant heat flux) are chosen because they are more realistic for the atmosphere. This configuration allows one to realize a variety of flow regimes from axisymmetric to highly irregular (Sukhanovskii et al., 2023; Vasiliev et al., 2023). Motivated by the problem of Arctic warming we examine how central cooling affects the structure and characteristics of the flow, which is similar to the typical atmospheric circulation. The mean meridional circulation (Fig. 1a) includes analogs of the polar cell at small radii, the weak Ferrel cell (which is seen only after averaging over zonal coordinate and time) at middle radii and the Hadley cell at the periphery. The shift of the heater from the sidewall leads to the formation of an additional cell to the right of the Hadley cell analog. This fourth cell provides anticyclonic circulation near the sidewall, which resembles easterly winds in the lower latitudes (Fig. 1b). In the upper layer, the analogs of the polar and Hadley cells transport the fluid with relatively large values of angular momentum to the smaller radii, providing formation of pronounced cyclonic zonal flows. A weak cell in the middle radii is the result of a train of baroclinic waves (Fig. 1c) that efficiently transport heat from the analog of the Hadley cell to the polar cell. To model Arctic warming, we vary the size and power of the central cooler for a fixed heat flux at the periphery. There are many issues associated with the Arctic amplification problem, but in the present study we limited ourselves to considering changes in the mean flow structure, mean heat transport, and baroclinic wave characteristics.

As non-dimensional governing parameters, following (Scolan and Read, 2017), we use the thermal Rossby number $Ro_T$, the Taylor number $Ta$, and the Ekman number $E$:

$$Ro_T = \frac{g\alpha h\Delta T}{\Omega^2 R^2}, \tag{1}$$

$$Ta = \frac{4\Omega^2 R^5}{h\nu^2}, \tag{2}$$

$$E = \frac{\nu}{\Omega h^2}, \tag{3}$$

where $g$ is the gravitational acceleration, $\alpha$ is the thermal expansion coefficient, $\Delta T$ is the temperature difference between heater and cooler, $\Omega$ is a rotation rate, $R$ is the radius of the layer, and $\nu$ is the kinematic viscosity.

Several remarks should be made regarding the thermal Rossby number. It is a key parameter used in the study of a rotating cylinder gap filled with a fluid and isothermal sidewalls. Here we consider a rotating shallow cylindrical layer with non-uniform heating and cooling at horizontal boundaries. Moreover, instead of a constant temperature (first type boundary conditions), a constant heat flux is applied (second type boundary conditions). Therefore, we provide values of $Ro_T$ for comparison with the results of other studies, but this should be done with caution.

## 3 Methods

### 3.1 Experiment

A detailed description of the laboratory model of general atmosphere circulation is given in (Sukhanovskii et al., 2023; Vasiliev et al., 2023). The experimental model is a tank of a square cross-section with a side $L = 700$ mm, and height $H = 200$ mm

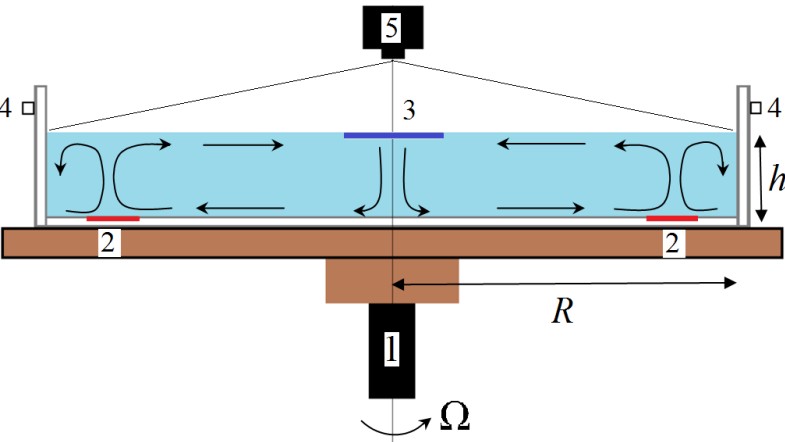

**Figure 2.** Scheme of the laboratory model, 1 - rotating table, 2 - rim heater, 3 - cooler, 4 - LED illumination, 5 - CCD camera. (Colour online)

(Fig. 2). The sidewall and bottom are made of Plexiglas with a thickness 20 mm. For the realization of the cylindrical layer the Plexiglas cylinder with a 3 mm wall and diameter $D = 690$ mm is inserted into the tank. The heater is a 25 mm wide circular strip of thin copper foil heated by an electric current. The distance from the cylindrical sidewall to the outer border of the heater is 40 mm. The heating power is controlled and kept constant during the experiment. The room temperature is kept constant by an air-conditioning system, and the cooling of the fluid is provided by the heat exchange with the surrounding air on the free surface, the central cooling system and some heat losses through the sidewall. The cooling system includes a thick (10 mm) copper disc with diameter $d = 56$ mm partially inserted into the upper layer of the fluid (about 2 mm). The upper surface of the copper disc is cooled by a thermoelectric (Peltier) cooler. To remove heat from the hot side of the thermoelectric cooler a radiator with a forced air circulation is used. For minimization of the impact of the air circulation, the cooling system is surrounded by an additional open box. Note that the size of the cooler, which is in direct contact with a fluid, is substantially less than the visible part of the cooling system (the white circle in the center of a Fig. 1c). The temperature of the cooler was measured by a copper–constantan thermocouple installed into the copper disc.

The experimental model is placed on a rotating horizontal table. The rotating table provides a uniform rotation in the angular velocity range $0.02 \leq \Omega \leq 0.30$ rad s$^{-1}$ (with accuracy of $\pm 0.001$ rad s$^{-1}$). The silicon oil PMS-5 (see Table 1) is used as the working fluid. In all the experiments presented, the depth of the fluid layer $h$ was 30 mm and the surface of the fluid was open. The temperature inside the fluid layer was measured at mid-height ($z = 15$ mm) and $R = 180$ mm by the copper–constantan thermocouple and used for the estimation of the mean temperature of the fluid. The main fluid properties and parameters of the experimental set-up are provided in Table 1. The direction of rotation in all experiments was clockwise.

Aluminum flakes are used to visualize the flow structure in the upper layer. The illumination of the tracers is provided by LED (light-emitting diode) strip placed on the perimeter of the experimental model above the fluid layer. The aluminum flakes are oriented along the flow, so they are bright when the flow is horizontal and dark when vertical motions are dominant. The recording was provided by 4 Mpx CCD camera Bobcat 2020 with 1 fps.

| Fluid properties | Symbol | Value | Units |
|---|---|---|---|
| Density | $\rho$ | 911 | kg m$^{-3}$ |
| Kinematic viscosity | $\nu$ | $5.2 \times 10^{-6}$ | m$^2$ s$^{-1}$ |
| Thermal diffusivity | $\kappa$ | $8.3 \times 10^{-8}$ | m$^2$ s$^{-1}$ |
| Thermal expansion coefficient | $\alpha$ | $9 \times 10^{-4}$ | K$^{-1}$ |
| Prandtl number | $Pr = \nu\,\kappa^{-1}$ | 62.7 | |
| | | | |
| Experimental set-up | | | |
| Layer radius | $R$ | 345 | mm |
| Layer depth | $h$ | 30 | mm |
| Heater width | $l$ | 25 | mm |
| Heater radius | $r_h$ | 293 | mm |
| Cooler radius | $r_c$ | 28 | mm |
| Heating power | $P_h$ | 123 | Wt |
| Cooling power | $P_c$ | $\approx 3$ | Wt |

**Table 1.** The main fluid properties and parameters of the experimental model.

It should be noted that the cooler in the experiment is relatively small. This was done deliberately to minimize the loss of angular momentum due to friction at the solid boundary (Evgrafova and Sukhanovskii, 2022) and intensification of vertical circulation by Ekman pumping. The cooler induces intensive descending flow in the central area, but the substantial difference in the heating and cooling areas leads to a drastic difference between cooling and heating power. This means that most of the cooling is provided by the heat exchange between the fluid and the air on the open surface.

### 3.2 Numerical simulation

The experiments carried out to study the flow structure at the upper level yielded valuable information, but they are not sufficient to understand all aspects of a complex system. An effective way to solve this problem is numerical simulation using a digital "twin" of the laboratory model. This has already been demonstrated with the mathematical model implemented by the freely distributed computational fluid dynamics package OpenFOAM v2106 (Vasiliev et al., 2023). In the present study, the mathematical model of the laboratory system implemented by the in-house CFD code $\sigma$Flow was used for numerical simulations.

The scheme of the numerical model is shown in Fig. 3. The model consists of a rotating fluid layer in the cylindrical cavity with local cooling and heating. Most of the characteristics of the numerical model are similar to the laboratory model, but some differences should be mentioned. In the experiment, cooling at the top surface is provided by ambient air circulation and is not uniform. In numerical simulations, we apply a constant heat flux at the free fluid surface. In order to check the influence

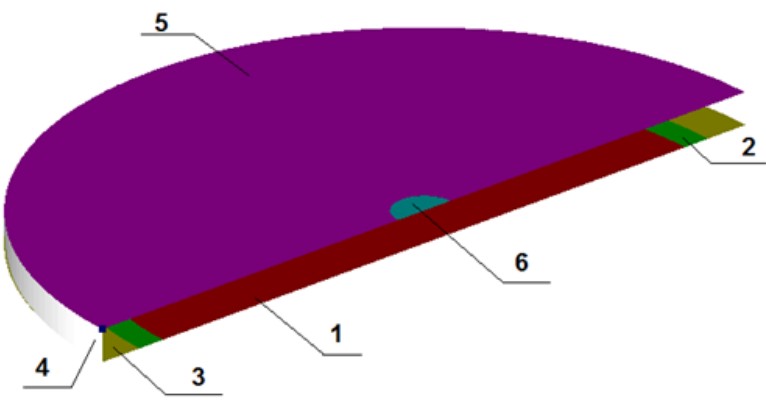

**Figure 3.** Geometry and boundaries of the computational domain. 1,3,4 - adiabatic walls, 2 - rim heater, 5 - free surface with constant heat flux, 6 - cooler. (Colour online)

of the solid boundary of the cooler, we use both non-slip and slip conditions for the surface of the cooler. Another important issue is the size of the cooler and its performance. To address this issue, we perform a series of numerical simulations with a
145 larger cooler (Table 3). The unsteady flow of an incompressible fluid is modeled in the Boussinesq approximation (in a rotating reference frame). The equations in a rotating reference frame are formulated in terms of the absolute velocity components.

*Boundary conditions* The boundary conditions are chosen to mimic the experimental apparatus. Non-slip conditions are imposed on all solid walls, including the cooler surface. A slip condition is imposed on the free boundary. Constant uniform heat fluxes are applied to the heater, cooler, and free surface, determined by the given net heat power of the corresponding
surface. The heat power of the free surface is equal to the difference between the heat powers of the heater and the cooler. The other surfaces are adiabatic. As an initial approximation for the velocity, the condition of solid body rotation at a given rotational speed is used. The initial uniform temperature of the fluid is equal to the reference temperature $T_0 = 20°C$.

*Discretization* The unstructured computational grid is constructed from several blocks with a structured hexagonal mesh. The mesh blocks distinguish the heater and cooler regions. A detailed description is given in the section on computational
verification.

The time step remained constant during the calculation and was set to 0.05 s. For the base mode with a rotation period T = 27 s, the maximum Courant number (CFL) calculated using the relative velocity does not exceed unity, and the volume average CFL = 0.085. The mean characteristics are obtained by averaging over time and along a uniform zonal direction after reaching the statistical steady state regime, which takes about 2000 c. The averaging time is at least 7600 s.
The numerical algorithm implemented in the CFD code $\sigma$Flow is based on the finite volume method for the unstructured mesh. The highlights of the algorithm are briefly listed below. For the spatial discretization, central differencing is used for the diffusion terms, and the convective terms of the momentum equation are approximated by a central second order difference scheme. A version of the Total Variation Diminish (TVD) scheme is used for the convective term in the heat energy transport equation. The numerical algorithm is based on a SIMPLE-like pressure correction procedure and a collocated grid array with

| Parameter | \|value\| | $p$ | GCI(fine), % | GCI(base), % | GCI(coarse), % |
|---|---|---|---|---|---|
| $T - T_0$ | $4.146\ \mathrm{K}$ | 1.288 | 1.2 | 2.4 | 5.4 |
| $k$ | $10^{-5}\ \mathrm{m^2\,s^{-2}}$ | 1.985 | 1.7 | 5.1 | 18 |
| $V_\phi$ | $0.00789\ \mathrm{m/s}$ | 1.718 | 1.7 | 4.3 | 12.8 |
| $q_t$ | $0.00087\ \mathrm{mK/s}$ | 1.991 | 1.2 | 3.5 | 12.5 |

**Table 2.** Estimation of discretization error.

Rhie-Chow interpolation. The unconditionally stable second order Crank-Nicolson method is used for the time integration. Both viscous and convective terms of the equation of motion are implicitly approximated. The system of linear algebraic equations for the pressure correction equation is solved using an algebraic multigrid solver.

*Verification* Verification of the simulation was performed by comparing the numerical results obtained on three different meshes. Verification calculations were performed for the basic mode with a rotation period $T = 27$ s, and heat power $\mathrm{Q_h} = 123$ Wt (heater), $\mathrm{Q_c}$ = -3 Wt (cooler), $\mathrm{Q_{fs}}$ = -120 Wt (free surface). Approximately the basic grid has the following spatial discretization: $N_r = 192$ nodes in radial direction with clustering of nodes to the heater and cooler, $N_\phi = 260$ nodes in tangential direction and $N_z = 40$ nodes in vertical direction with clustering factor to the boundaries 1.05. The total number of control volumes is $N_r \times N_\phi \times N_z = 2.0$ million cells. The fine mesh has the following discretization: $N_r = 375$, $N_\phi = 480$, $N_z = 60$, total number of control volumes 10 million. The coarse grid discretization is as follows: $N_r = 90$, $N_\phi = 100$, $N_z = 30$, total number of control volumes 0.285 million cells.

Using the numerical solution on three grids we can perform a procedure for the estimation of grid convergence and discretization error. The discretization error is calculated by the algorithm described in (Celik et al., 2008). Table 2 presents the results of the grid convergence analysis for the radial distributions of temperature difference $T - T_0$, turbulent kinetic energy $k$, relative tangential velocity $V_\phi$ and turbulent radial heat flux $q_t$. The obtained apparent order of spatial approximation $p$ is found to be close to the formal second order of accuracy, with the exception for the temperature field. This agreement is an indication of the grids being in the asymptotic range. The difference between the apparent and formal orders for the temperature field is most likely due to the upwind TVD scheme for convective terms. The values of the grid convergence indices are normalized to the values indicated as \|value\| in the Table 2. The fine-grid convergence index GCI, determined by comparing the results obtained for the detailed and basic grids, does not exceed 5%. The presented data confirm that the numerical accuracy of the base mesh is within acceptable limits for CFD simulation.

## 4 Results

### 4.1 Flow structure

Earlier in (Sukhanovskii et al., 2023) it was shown that the atmospheric regime, in which the mean flow and baroclinic waves are similar to those in the real atmosphere, i.e. strongly non-stationary waves with main modes $m = 2-8$ (see mode decomposition

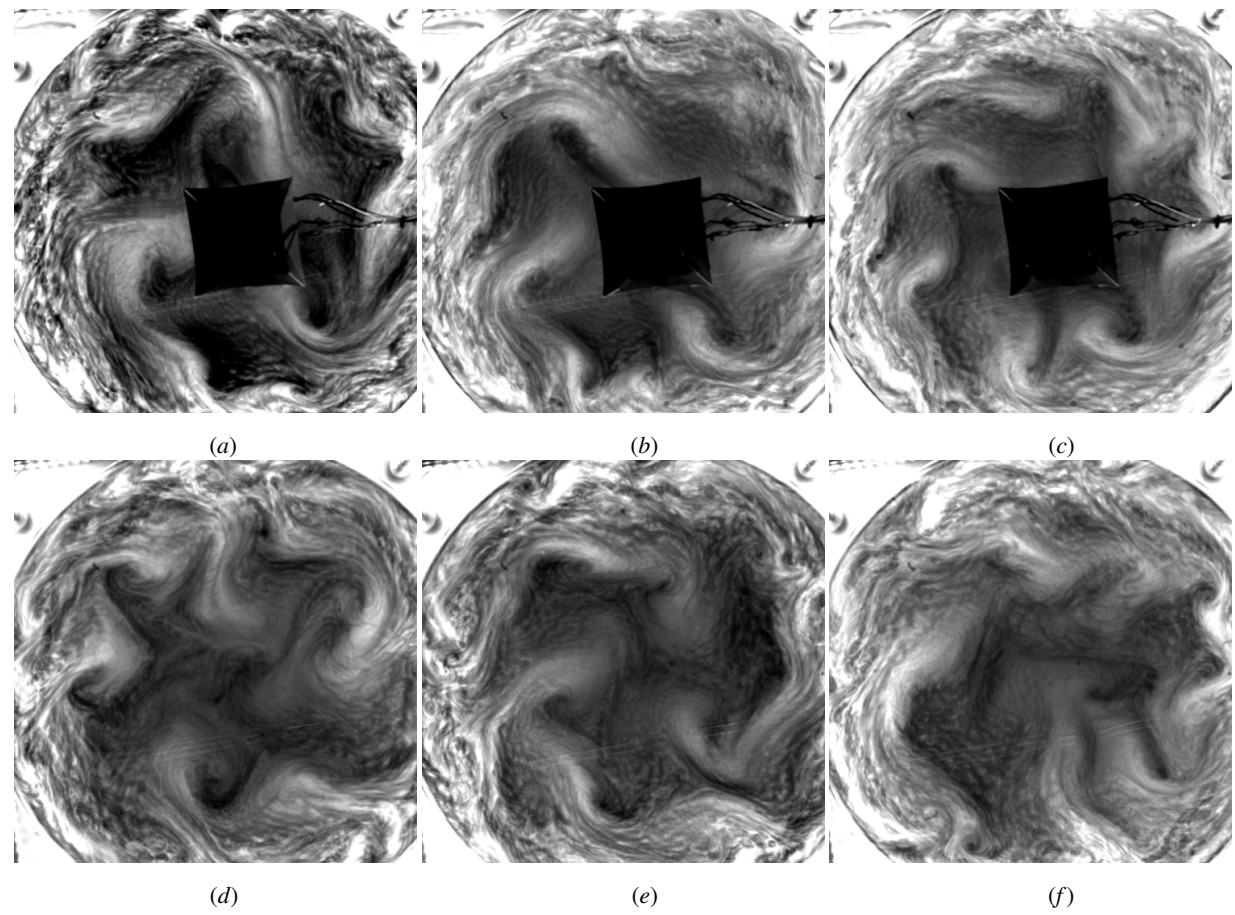

(a)                     (b)                   (c)

(d)                     (e)                   (f)

**Figure 4.** Examples of the flow structure in the atmospheric regime (cases 3 and 4). Upper panel (a-c) with cooler, lower panel (d-f) without central cooler. Experiment.(Colour online)

analysis in (Lembo et al., 2019, 2022)), can be realized only in a short interval of the main parameters. We have chosen this regime to study Arctic warming using laboratory and numerical modeling. To examine the role of localized cooling, we vary the size of the cooler and consider three cases: a large cooler ($r_c = 46$ mm, only in the numerical simulation), a small cooler ($r_c = 28$ mm) and uniform cooling at the free surface, without a local cooler. The heat flux for the large cooler was the same as for the small one, so the power of the cooler increased proportionally to its area from 3 Wt to 8 Wt. The net cooling power, including the free surface, is the same for all cases considered – 123 Wt. In order to better understand the role of Ekman pumping (due to the non-slip condition in the cooling area), we perform numerical simulations in the "atmospheric" regime for the central cooling of the large size ($r_c = 46$ mm) with non-slip boundary conditions. The main parameters of the experiments and numerical simulations are presented in Table 3.

Examples of the flow structure for the atmospheric regime with and without central cooling in experiments (case 3 and 4,Table 3) are shown in Fig. 4. For both cases the baroclinic waves in middle radii are not regular and are characterized by

| Case | $\Omega$, rad s$^{-1}$ | $Q_c$ | $r_c$, mm | $\Delta T$ | $Ro_T$ | $Ta$ | $E$ | $BC_{cooler}$ | $real.$ |
|------|------|------|------|------|------|------|------|------|------|
| 1 | 0.37 | 8 | 46 | 20.3 | 0.34 | $3.6 \times 10^9$ | 0.015 | slip | num |
| 2 | 0.37 | 8 | 46 | 24.3 | 0.41 | $3.6 \times 10^9$ | 0.015 | non-slip | num |
| 3 | 0.37 | 3 | 28 | 24.1 | 0.4 | $3.6 \times 10^9$ | 0.015 | non-slip | exp/num |
| 4 | 0.37 | - | - | 17.1 | 0.29 | $3.6 \times 10^9$ | 0.015 | no cooler | exp/num |

**Table 3.** Main parameters of experiments and numerical simulations

strong temporal and spatial variations. The difference in the flow structure between two configurations is not obvious. Typical flow structures for all four cases, considered in numerical simulations are shown in Fig. 5. Surprisingly, a significant increase in size and cooling power, and a change from non-slip to slip boundary conditions (Fig. 5a,b), does not lead to a noticeable change in the flow structure. In general, the instantaneous flows observed in the atmospheric regime are rather irregular and
similar for all configurations under study.

Qualitative observations of the flow structure in the upper layer provide only partial information about the flow. Using numerical modeling, we can reconstruct the flow structure in the vertical cross section and show quantitative changes between different configurations. One of the specific problems of laboratory modeling of general atmospheric circulation is the realization of polar cooling. Experimental realization of contactless cooling is a complex technical problem, and usually a solid heat
exchanger is used. It is either a cylindrical inner wall or a disk cooler on the upper surface. Both configurations are not realistic and can lead to noticeable qualitative differences between the flow in a laboratory model and in the real atmosphere. To check the influence of the solid cover on the flow formation, we consider the large cooler with a slip boundary condition, which is more realistic.

The mean flows in a vertical cross section (meridional circulation) for different configurations are shown in Fig. 6. There
is a significant difference in the flow structure between two configurations with the large cooler but under different boundary conditions (Fig. 6a,b). The cooler with non-slip boundary condition has a viscous boundary layer (Ekman layer), which leads to an additional circulation caused by the Ekman pumping. Analysis of the results obtained indicates that the Ekman pumping provides an intensive downward flow near the axis of rotation, which is crucial for the structure of the laboratory polar cell. If we turn off the Ekman pumping using slip boundary conditions, then the polar cell analog becomes substantially weaker and
changes its shape. The polar cell moves up and closer to the middle radii. A decrease in the size of the non-slip cooler results in a decrease of the size of the polar cell but the overall structure remains unchanged. If we consider the case without cooler (uniform heat flux at the free surface), then the polar cell structure is similar to the case with the large slip cooler but the polar cell analog becomes even weaker, shifts closer to the middle radii, and is located in the upper layer. Based on the results, we can conclude that the size and boundary conditions at the cooler surface play a key role for the structure and intensity of the
polar cell analog. The slip cooler provides the overall structure, which is qualitatively more similar to the Earth's meridional circulation, with an intense Hadley cell, a much weaker Ferrel cell, and an even weaker polar cell (Dima and Wallace, 2003).

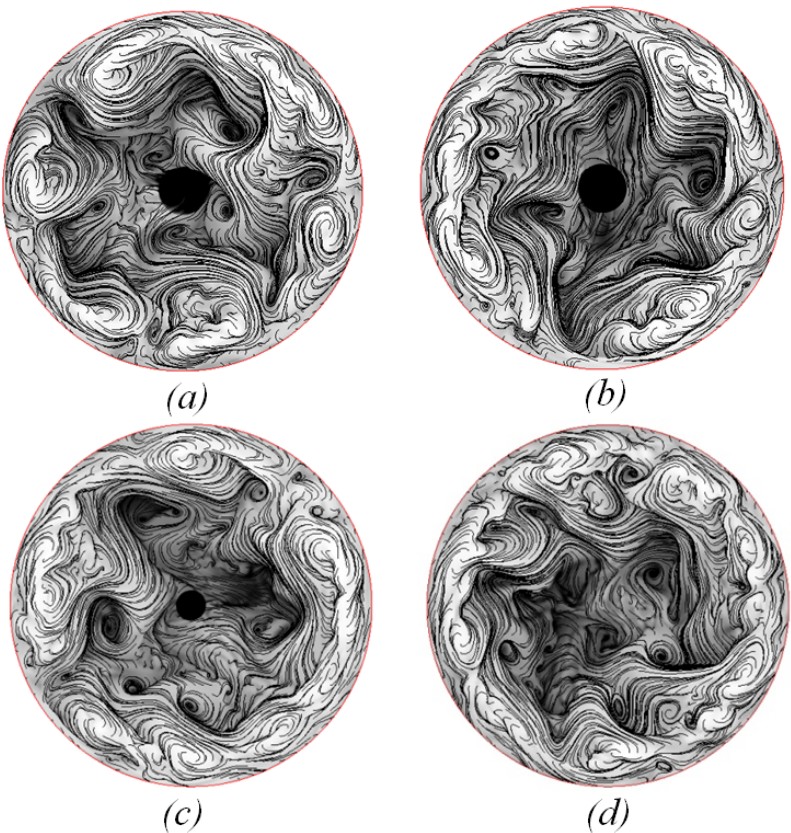

**Figure 5.** Typical flow structure at the top layer for different configurations, the lines are trajectories of fluid particles (streamlines) and the shading characterizes temperature distribution (white – hot, black – cold), the black circle in the middle – the cooler. a – large slip cooler, b – large no-slip cooler, c – small no-slip cooler, d – no cooler. Numerical simulation.(Colour online)

Another important issue related to Arctic warming is the change in the zonal velocity distribution. Meridional cells transfer the angular momentum and provide the formation of the differential rotation (zonal flows). We can expect that the transformation of the polar cell analog achieved by varying the size and intensity of local cooling would lead to the change of the zonal flow structure and intensity. Fig. 7 illustrates the influence of the laboratory Arctic warming on the mean zonal velocity (averaged over time and zonal coordinate) for different cases. In Fig. 7a we see the zonal velocity distribution for the more realistic case with a large localized slip cooler. There are analogs of easterly winds in the large radii (laboratory low latitudes) and westerly winds in the middle radii (laboratory mid-latitudes). The response on the laboratory Arctic warming (turning off the localized cooling) is shown in Fig. 7b. The main deviations are observed in the small radii due to a significant change in the structure of the laboratory polar cell. For the quantitative comparison we plot in Fig. 7c the zonal velocity profiles near the surface for all considered cases (the case with a polar heater at the bottom is described in Appendix). The laboratory Arctic

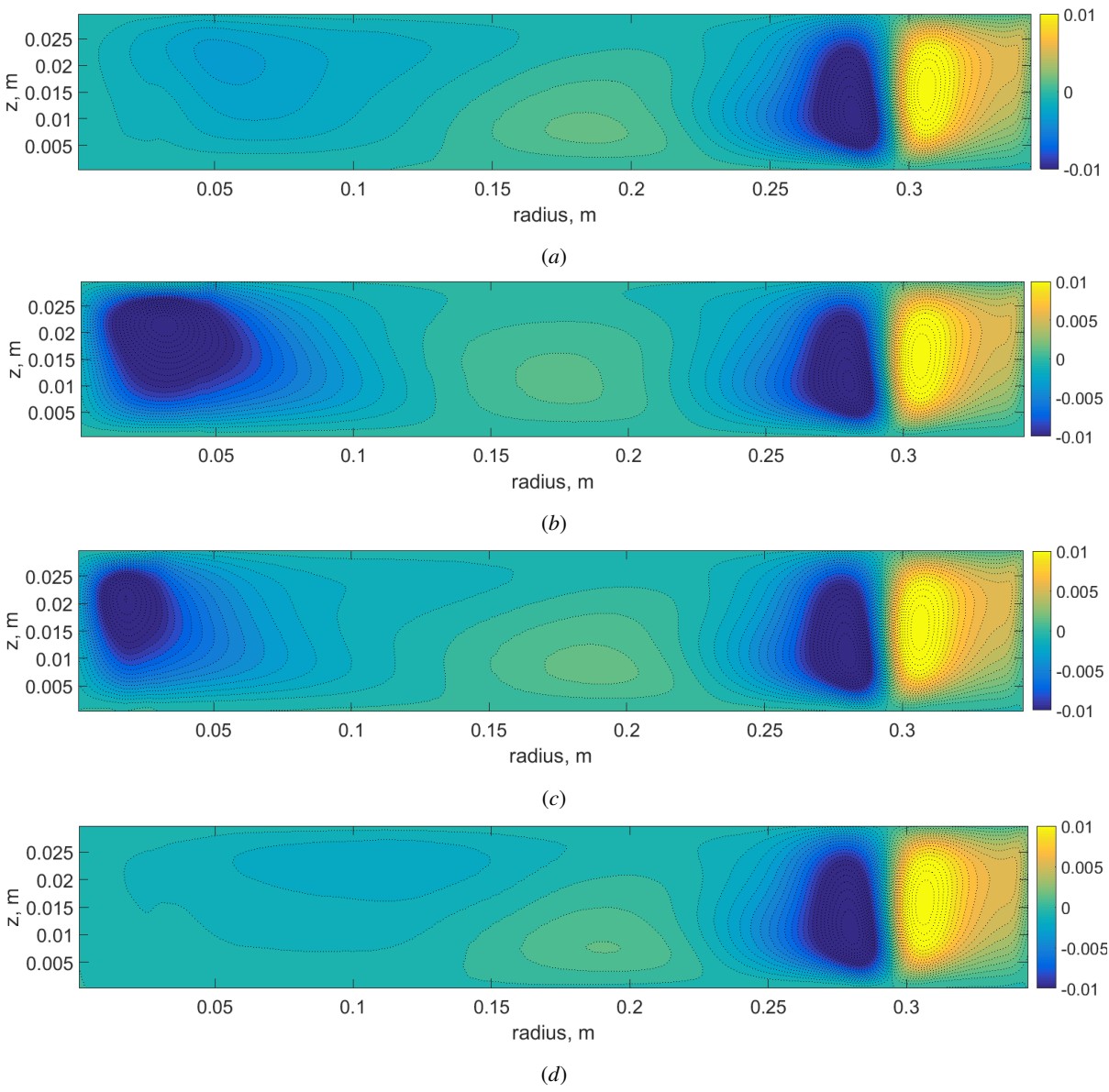

**Figure 6.** Mean meridional circulation (stream function) for different cases, a – case 1 (large slip cooler), b – case 2 (large non-slip cooler) , c – case 3 (small non-slip cooler), d – case 4 (no cooler). Numerical simulation. (Colour online)

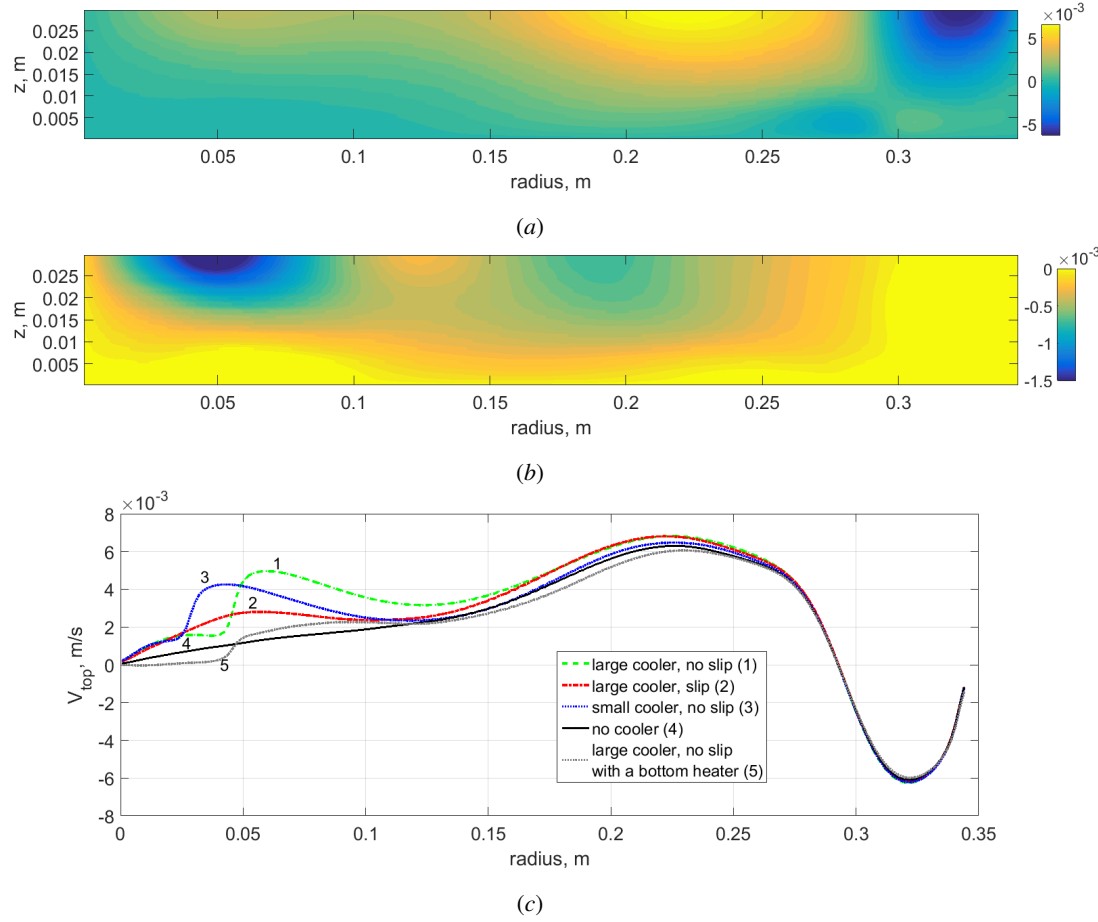

**Figure 7.** (a) – mean vertical field of relative zonal velocity $V_{lcs}$ in case of a large cooler with a slip condition (in a rotating frame); (b) – change of zonal velocity caused by laboratory Arctic warming $\Delta V = V_{nc} - V_{lcs}$ ($V_{nc}$ - correspond to the case without cooling); (c) – profiles of zonal velocity near the surface at $z = 0.028$ m, for all considered configurations. Atmospheric regime, velocity in m/s. Numerical simulation. (Colour online)

warming leads to the significant decrease of zonal velocity (about 60%) in the laboratory polar region and weaker decrease (less than 10%) in the laboratory westerly winds in the middle radii.

The next question concerns the influence of localized cooling on the characteristics of baroclinic waves that provide the radial

transport of heat and angular momentum. To illustrate the intensity and location of baroclinic waves, we present distributions of the mean energy of radial velocity fluctuations $E_{bw} = \langle u_r^2 \rangle_{\phi,t}$ in a vertical cross section for the case with a large cooler and slip conditions, which we consider as more realistic (Fig. 8a) and the changing of the energy of fluctuations after turning off the cooling (Fig. 8b). In all cases considered (large cooler with slip and non-slip conditions, small non-slip cooler, no cooler) the location of baroclinic waves and their energy distribution are similar to the one shown in Fig. 8a. The baroclinic waves are

formed in the upper part of the layer in the middle radii (laboratory mid-latitudes) with maximum intensity at $r \approx 0.22$m. The

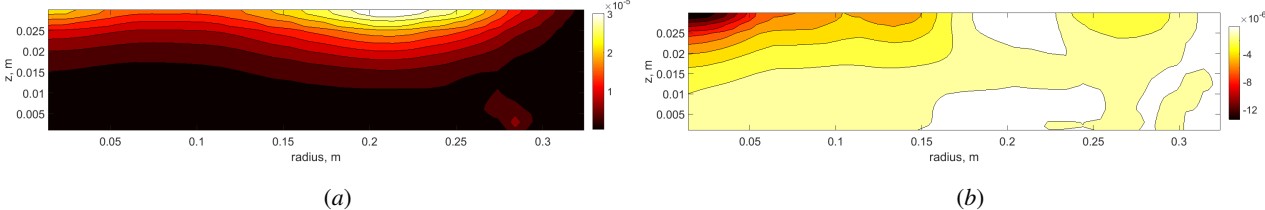

**Figure 8.** (a) – vertical field of energy of radial velocity fluctuations in case of a large cooler with slip condition, (b) – change in energy of radial velocity fluctuations after turning off the cooling. Numerical simulation. (Colour online)

turning off the cooling results mainly in decreasing of fluctuations in the upper layer near the axis (Fig. 8b). The profiles of $E_{bw}$ at the top of the layer are shown in Fig. 9a. There is a weak response of baroclinic wave activity in the middle radii to changes in the central cooling. The substantial differences in wave energy are seen only in the small radii, where turning off the cooling leads to the flow restructuring. In the case of large slip cooler there is a second maximum of $E_{bw}$, non-slip cooler results in a sharp decrease of $E_{bw}$ and in the case without localized cooler there is a monotonic decrease of $E_{bw}$.

The baroclinic waves in the atmospheric regime are the superposition of different wave modes in the zonal direction, and their energy can be estimated by Fourier decomposition (see e.g. (Sukhanovskii et al., 2023)). The energy distributions of main baroclinic modes averaged over area of maximal baroclinic wave activity (from $r = 0.19$ m to $r = 0.23$ m) are shown in Fig. 9b. As in the real atmosphere, modes $m = 4-8$ contain most of the baroclinic wave energy. There are noticeable deviations of the energy of individual modes for different cases. The temporal behavior of various baroclinic modes is rather complex and non-periodic (Sukhanovskii et al., 2023), and although the simulation time (more than 500 rotation periods) significantly exceeds the characteristic wave time scale (about 5-7 rotation periods), it is not sufficient to achieve convergence of the average energy values of individual modes. Thus, we can not separate the mode variations caused by changes in polar cooling and due to internal processes. To answer this question we need much longer numerical simulations.

## 4.2 Heat transfer and variation of the mean vertical temperature field

The main function of the large-scale meridional circulation and mid-latitude baroclinic waves is to transfer heat from the equator to the polar region. Fig. 10a shows the distribution of the total mean radial heat flux (averaged over zonal coordinate, height and time) along the radius. In the quasi-stationary state, the total heat flux is directed toward the center (polar region) and decreases monotonically due to cooling at the free surface and by the local cooler. The total heat flux can be divided into two parts, mean and pulsating (Fig. 10b), provided by the mean circulation and mean temperature distribution, and pulsations of velocity and temperature:

$$q_{full}(r) = 2\pi r(\langle U_r T \rangle_{\phi,h} + \langle u_r T' \rangle_{\phi,h,t}) \tag{4}$$

, where $U_r$,$T$ are the mean (over time) radial velocity and temperature, and $u_r$,$T'$ are the pulsations of radial velocity and temperature.

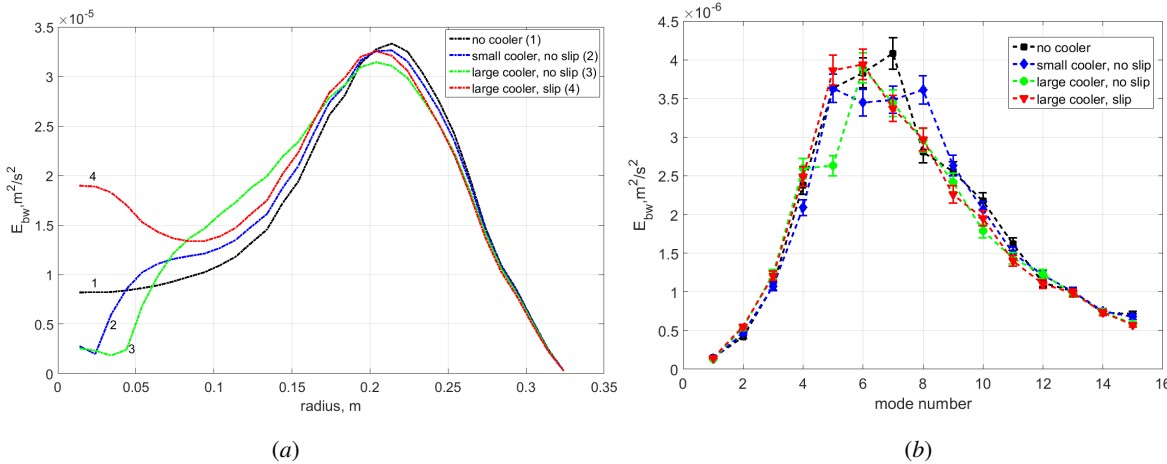

**Figure 9.** (a) – profiles of energy of radial velocity fluctuations at the top of the layer, (b) – energy of different modes of radial velocity fluctuations at the top of the layer, averaged over interval from $r = 0.19$ m to $r = 0.23$ m (area of maximal baroclinic wave activity). The errorbars characterize 95% confidence interval for the mean values of energy of separate modes. Different cooling configurations in the atmospheric regime.(Colour online)

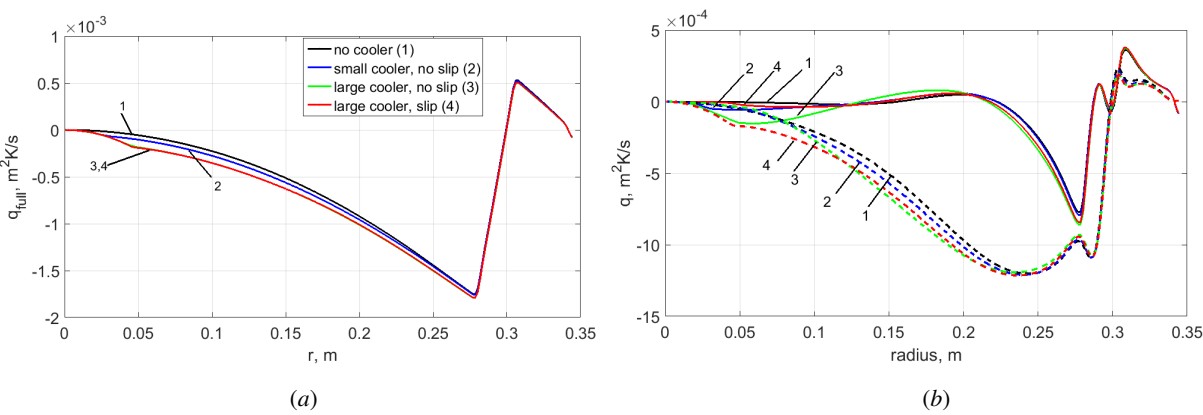

**Figure 10.** (a) – total radial heat flux (integrated over the zonal coordinate), averaged over time and height, black line (1) – without cooling, blue line (2) – small cooler, green line (3) – large cooler, non-slip condition, red line (4) – large cooler, slip condition; (b) – mean (solid lines) and pulsating (dotted lines) parts of the total heat flux (same legend as in the left panel). Atmospheric regime. Numerical simulation. (Colour online)

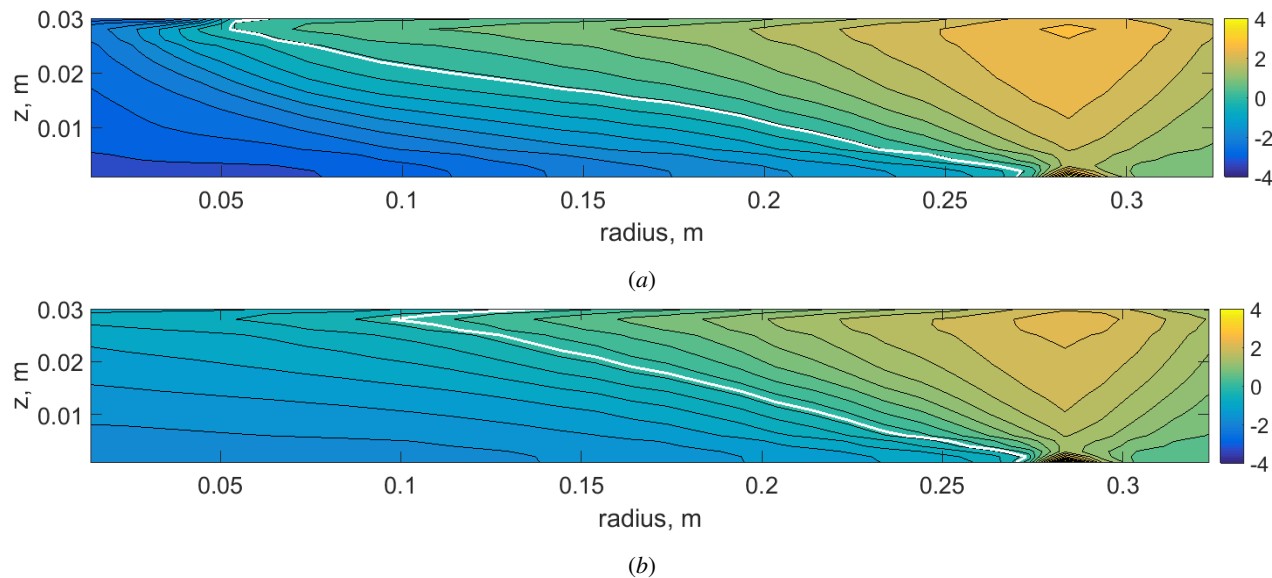

**Figure 11.** Mean vertical temperature fields (averaged over zonal coordinate and time). (a) – with large cooler and non-slip condition ($T_{lc}$), (b) – without cooler ($T_{nc}$). The mean temperature of the fluid ($T_0 = 293$ K) is subtracted. The solid white line shows zero isotherm. Atmospheric regime. Numerical simulation. (Colour online)

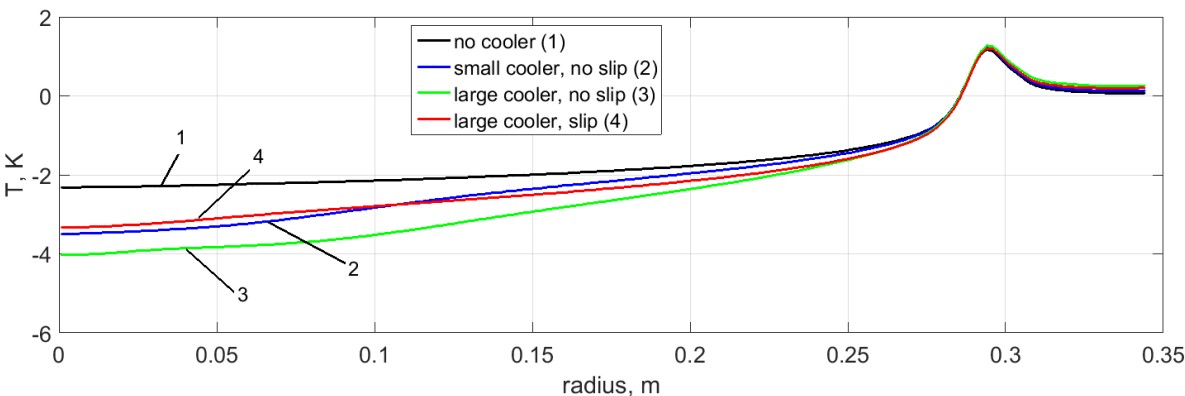

**Figure 12.** Mean temperature profiles in the lower layer ($z = 0.002$ m), black line (1) – without cooling, blue line (2) – small cooler, green line (3) – large cooler, non-slip condition, red line (4) – large cooler, slip condition. The mean temperature of the fluid ($T_0 = 293$ K) is subtracted. The solid white line shows zero isotherm. Atmospheric regime. Numerical simulation. (Colour online)

It is obvious that the pulsating part of the heat flux plays the key role in the heat transfer towards the polar region. Only in the case of a large non-slip cooler the mean heat flux (solid green line in Fig. 10b) is dominant in the polar region due to the effective suppression of the wave motion. The profiles of the mean (over height) heat flux for all cases are quite similar, because the variations of the boundary conditions are relatively weak, except in the region of the localized cooler. The heat flux in the heating region is strictly the same, and the variations of the heat flux at the upper surface are only noticeable in

the localized cooler zone. However, the distribution of the heat flux (which is mostly convective) along the vertical coordinate depends strongly on the flow structure. According to this, we can expect substantial spatial variations of the heat flux and, as a consequence, of the mean temperature distribution change, due to the remarkable transformation of the polar cell, which we described earlier. To check this assumption, we compare the mean vertical temperature fields for substantially different polar cell structures, namely, for the cases with a large non-slip cooler and without a cooler (Fig. 11). In fact, there is a noticeable

change in the temperature distribution, mainly in the central region and in the lower layer. For practical applications (e.g. weather forecasting) the surface temperature is one of the most important parameters. The mean temperature profiles near the bottom, are shown in Fig. 12. The main result is that the transition from non-uniform to uniform cooling conditions leads to an increase in temperature near the bottom, up to the heating region. Note that for all cases considered, the mean temperature of the fluid ($T_0 = 293$ K) and the total heating and cooling power are the same. The only difference is the distribution of the

cooling flux on the top surface. It is either non-uniform in the cases with a localized cooler or uniform in the case without a localized cooler. Noticeable changes in the temperature distribution can be better seen by plotting the temperature difference for different cases (Fig. 13)a,b. The removal of the local cooler leads to a significant transformation of the mean temperature field. The central region and most of the lower layer become warmer, while most of the upper layer and the peripheral (equatorial) part of the lower layer become colder. For the non-slip cooler, the temperature trends are stronger due to Ekman pumping, but

for the slip cooler, which is a better approximation of real atmospheric conditions, the effect is also substantial.

    To compare the numerical results with experimental data, temperature measurements for two cases (with and without localized cooling) were made by an array of thermocouples (16 thermocouples) in the bottom layer ($z = 5$ mm). During the open discussion, one of the referees commented that Arctic sea-ice loss would warm the bottom rather than reduce cooling in the upper layer. To address this comment, we performed numerical simulation with a large non-slip cooler at the top and

295 a large central heater at the bottom of equal power and size. More details of this simulation are presented in Appendix. The temperature difference profiles for numerical simulations and experiment at $z = 5$ mm are shown in Fig. 13c. As we can see, all temperature profiles show a significant temperature increase in the bottom layer without localized central cooling. Adding of the bottom heating and turning off the top cooler results in close temperature profiles, except for the central area above the heater. We can conclude that for the laboratory system the overall cooling in the central area plays crucial role for the

300 lower layer temperature distribution. We also note that the experimental points obtained in case of the small cooler are closer for the numerical simulation with a large cooler. We assume that the main source of this discrepancy is underestimation of the cooling power in the experiment. Qualitatively, the temperature trends presented are very similar to those obtained by the re-analysis (Screen and Simmonds, 2010).

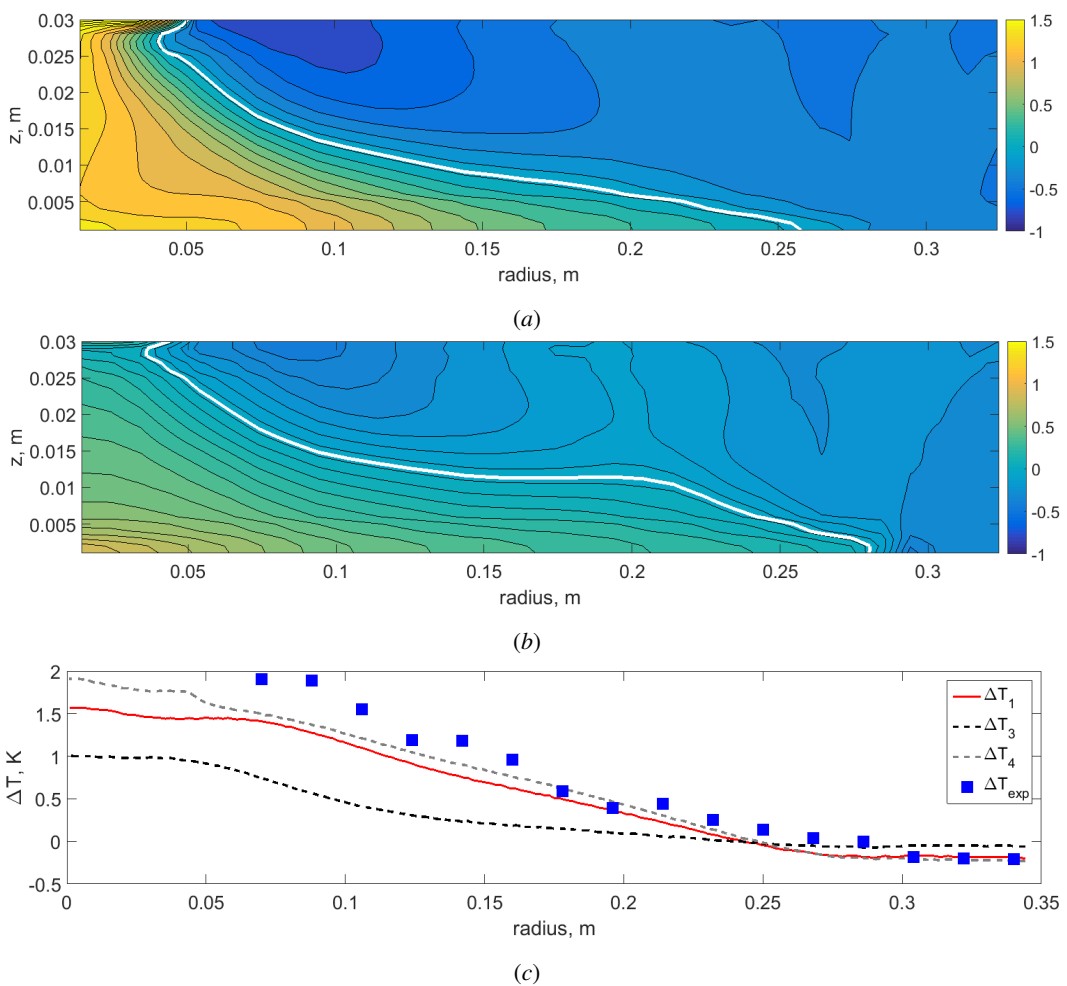

**Figure 13.** (a) – temperature difference $\Delta T_1 = T_{nc} - T_{lc}$, where $T_{nc}$ – mean vertical temperature field without cooler and $T_{lc}$ – with large non-slip cooler (numerical simulation); (b) – temperature difference $\Delta T_2 = T_{nc} - T_{lcs}$, where $T_{lcs}$ – mean vertical temperature field with large slip cooler (numerical simulation); (c) – profiles of temperature difference at $z = 5$ mm, red solid line – $\Delta T_1$, black dotted line – $\Delta T_3 = T_{nc} - T_c$, where $T_c$ – mean vertical temperature field with a small non-slip cooler, gray dotted line – $\Delta T_4 = T_{lcph} - T_{lc}$, where $T_{lcph}$ – mean vertical temperature field with a large non-slip cooler at the top and large polar heater of the same power at the bottom, blue squares – $\Delta T_3^{exp}$, experimental measurements by array of thermocouples. Atmospheric regime. (Colour online)

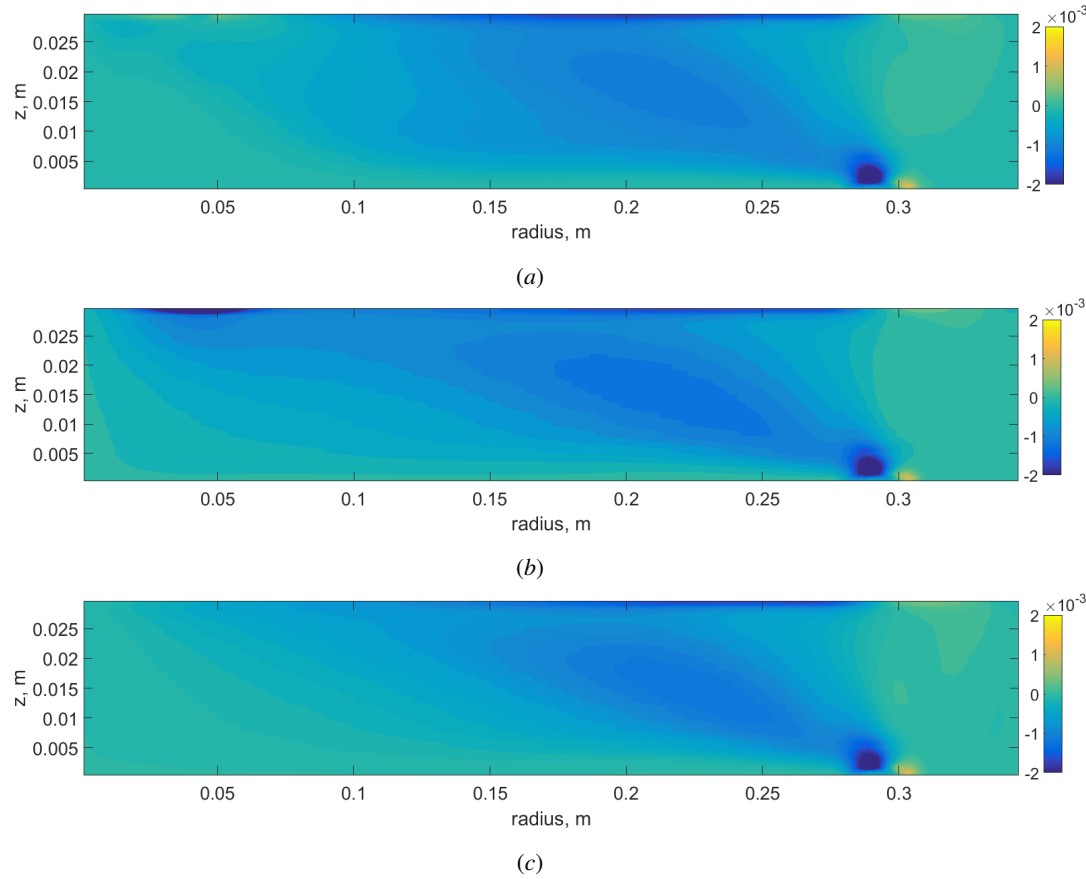

**Figure 14.** Pulsating part of the radial heat flux $\langle u_r T' \rangle_{\phi,t}$, in mK/s. (a) – regime with large non-slip cooler, (b) – regime with large slip cooler, (c) – regime without cooler. Atmospheric regime. Numerical simulation. (Colour online)

To explain this remarkable transformation of the mean temperature field, it is necessary to analyze the spatial distribution of heat flux. The vertical fields of the pulsating and mean parts of the radial heat flux are shown in Fig. 14 and Fig. 15. As we can see the spatial structure of the pulsating part of the heat flux (mainly provided by baroclinic waves) is similar for all cases. The pulsating heat flux transports heat to the upper layer and then to the center. In the lower part of the layer the wave motions are damped by viscous friction and the heat flux near the bottom is mainly provided by the mean circulation (Fig. 15). The intensive polar cell in the case of a large non-slip cooler provides a cold fluid flux towards the periphery near the bottom (negative heat flux), which cools the lower part of the layer (Fig. 15a). The transition from non-slip to slip boundary conditions at the localized cooler (switching off the Ekman pumping) leads to remarkable decrease in the negative heat flux near the bottom (Fig. 15b) and consequently to an increase of the temperature (Fig. 13). The next transition from the large slip cooler to the uniform cooling results in a further decrease of the negative heat flux (Fig. 15c). For quantitative comparison of the mean and pulsating heat flux near the bottom, the corresponding profiles are shown in Fig. 16.

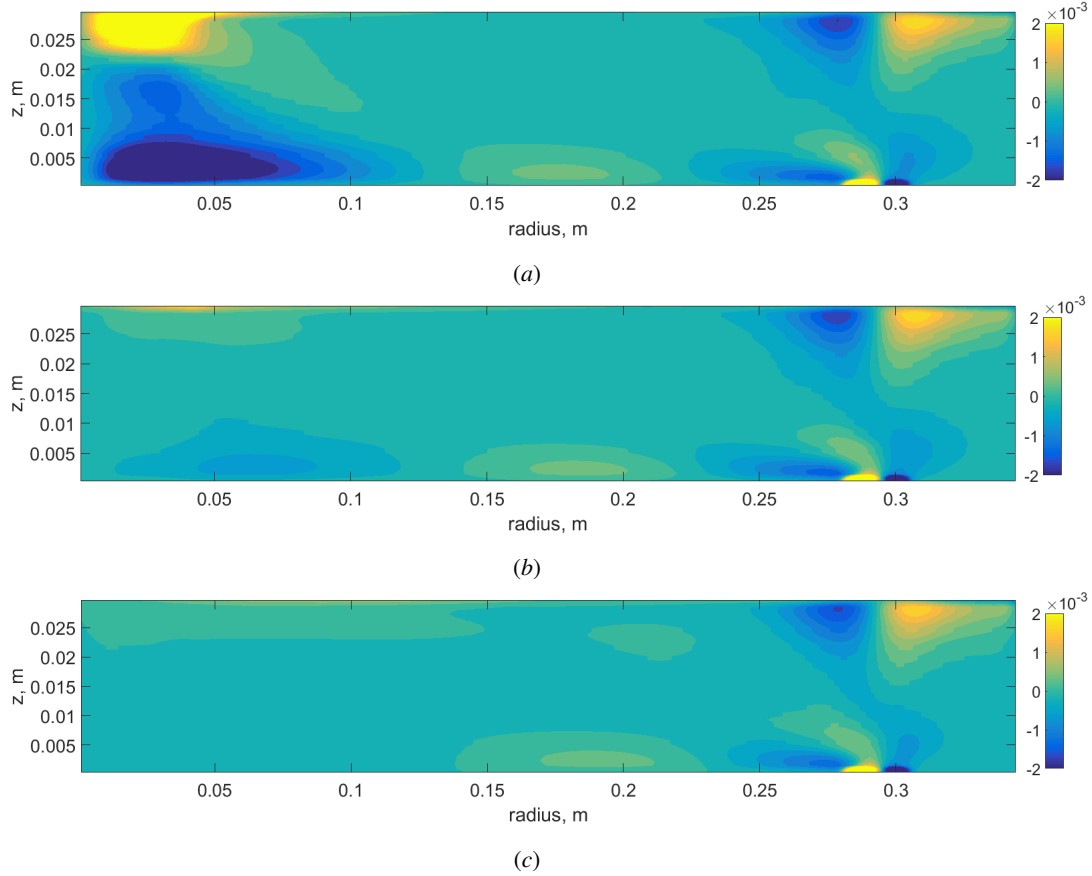

**Figure 15.** Mean part of the radial heat flux $\langle U_r T \rangle_\phi$, in $\mathrm{mK/s}$. (a) – regime with large non-slip cooler, (b) – regime with large slip cooler, (c) – regime without cooler. Atmospheric regime. Numerical simulation. (Colour online)

## 5   Conclusions

The results of experimental and numerical modeling of Arctic warming in a laboratory dishpan configuration are presented. Arctic warming is reproduced by varying the size of the local cooler in the "atmospheric" regime, when the structure of the flow is similar to the general atmospheric circulation. Namely, the meridional circulation consists of three cells, laboratory analogs of Hadley cell, Ferrel cell and polar cell. The baroclinic waves in this regime are strongly non-stationary with the main modes $m = 4 - 8$.

The laboratory Arctic warming results in a relatively weak response of the meridional and zonal circulation except in the polar region, where the polar cell analog becomes weaker, shifts closer to the middle radii, and is mainly located in the upper layer. The structure of analogs of Hadley and Ferrel cells is the same for all considered configurations. In the extreme case (without polar cooling) the reduce of the velocity of the zonal flow (analog of westerly wind) was about 10%. This relatively small decreasing of zonal flow velocity is in a good agreement with results of modeling of Arctic sea-ice loss (Blackport and

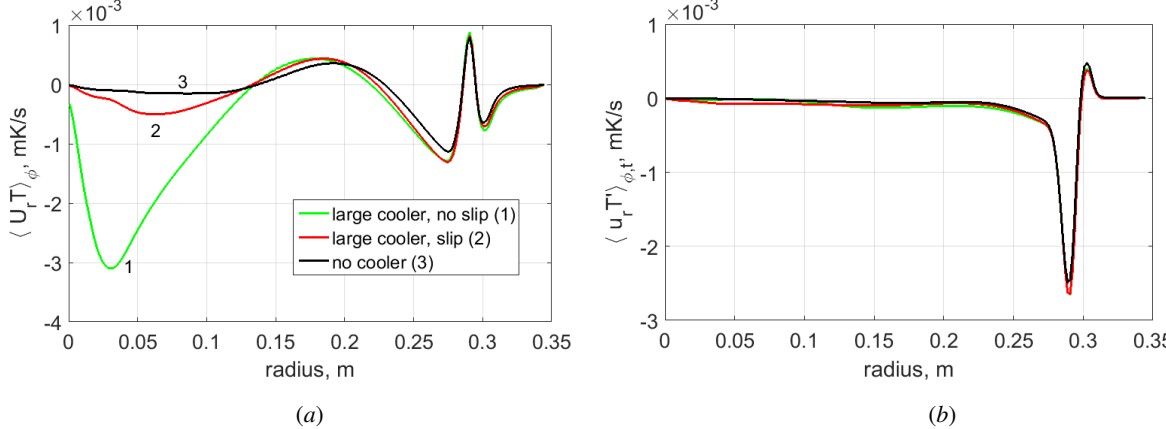

**Figure 16.** Profiles of the mean and pulsating parts of the radial heat flux near the bottom, at $z = 0.002$ m. (a) $- \langle U_r T \rangle_\phi$; (b) $- \langle u_r T' \rangle_{\phi,t}$. Green line (1) – large cooler, non-slip condition, red line (2) – large cooler, slip condition, black line (3) – without cooling. "Atmospheric" regime. Numerical simulation. (Colour online)

Screen, 2020; Smith et al., 2022; Ye et al., 2024). The baroclinic waves in the mid-latitudes are crucial for the heat and mass transfer and strong temporal variations. We observed only a weak response of the baroclinic wave activity to the laboratory Arctic warming, which is also in a good agreement with full scale numerical modeling (Blackport and Screen, 2020; Ye et al., 2024). We assume that the main reason of the weak effect of the laboratory Arctic warming on the laboratory mid- and low-
330 latitudes is the relatively small overall cooling power in the laboratory polar area (which occupies only about 2% of the surface). Even in the case of the large cooler it is less than 10% of the total cooling power at the surface.

The main result of laboratory Arctic warming is a noticeable transformation of the mean temperature field, namely, the polar region and most part of the lower layer become warmer, while most of the upper layer and the peripheral (equatorial) part of the lower layer become colder. The nature of this phenomenon in the system under consideration is described on the base
of our numerical data. It is closely related to the change in the radial heat fluxes. The baroclinic waves transport heat to the upper layer and then to the center. In the lower part of the layer the wave motions are damped by viscous friction and the heat flux near the bottom is mainly provided by the mean circulation. The removal of local cooling leads to a weakening of the analog of polar cell and a significant decrease in the negative heat flux near the bottom, which inevitably leads to an increase in temperature.

The results of laboratory modeling cannot be directly extrapolated to the real atmosphere, but the obvious similarity between large-scale laboratory circulation and general atmospheric circulation, including mid-latitude wave activity give some support for consideration of the described scenario as one of the plausible explanations for Arctic warming amplification.

One of the specific problems in laboratory modeling of general atmospheric circulation is the realization of polar cooling. Usually this is either a cylindrical inner wall or a disc cooler on the upper surface. The Ekman pumping at the cooler surface
can lead to noticeable qualitative differences between the flow in a laboratory model and in the real atmosphere. It is found that

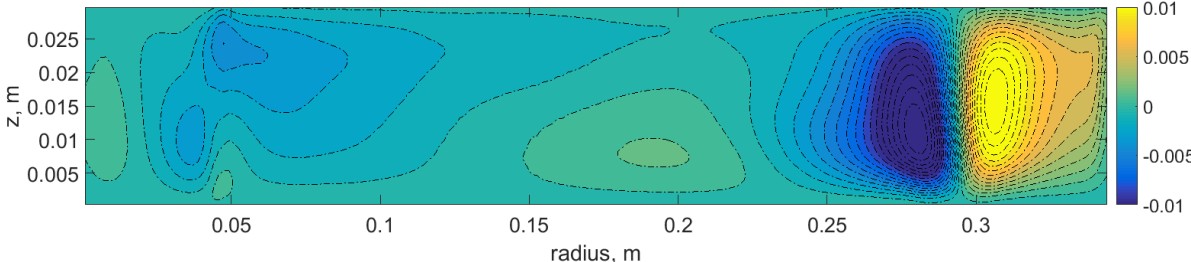

**Figure A1.** Mean meridional circulation (stream function) for the case with the local non-slip cooler and the heater at the bottom of the same power. Numerical simulation. (Colour online)

Ekman pumping results in a strong descending updraft near the axis of rotation, which determines the structure of the central meridional cell. In the case of the slip cooler, which better simulates real atmospheric conditions, the polar cell analog is much weaker and located mainly in the upper layer, closer to the central radii. We can conclude that the size and boundary conditions at the surface of the cooler have a strong influence on the structure and intensity of the polar cell analog.

## Appendix A: Case with a polar heater

During the open discussion, one of the referees (Tim Woolings) made reasonable comment that Arctic sea-ice loss would warm the bottom rather than reduce cooling in the upper layer. To address this comment, we performed numerical simulation with a large non-slip cooler at the top and a large central heater at the bottom of equal power and size, without changing other parameters (such as grid resolution, time step, etc.). The main outcome of this simulation is in the frame of the results described in the paper. The bottom heating has strong influence on the structure of laboratory polar cell (Fig. A1) but there is a weak response in the laboratory mid- and low-latitudes. The same conclusion is for the zonal flow and baroclinic wave activity (Fig. A2 and Fig. A3). We expected stronger difference in the mean temperature field, but here we also see similar trends (Fig. A4a). The result of turning off the cooling and adding of the heating of the same power is very close, except the laboratory polar area (Fig. A4b). We can conclude that for the laboratory system the overall cooling in the central area plays crucial role for the lower layer temperature distribution.

*Author contributions.* All authors contribute equally.

*Competing interests.* The authors report no conflict of interest.

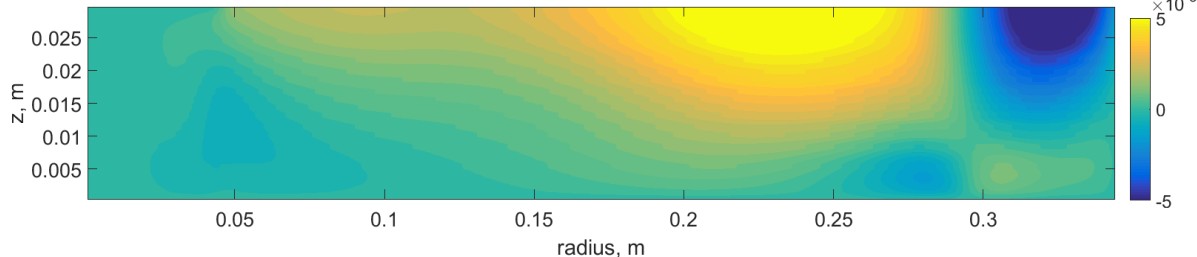

**Figure A2.** Mean vertical field of relative zonal velocity $V$ (in a rotating frame) in case with the polar heating. Numerical simulation.(Colour online)

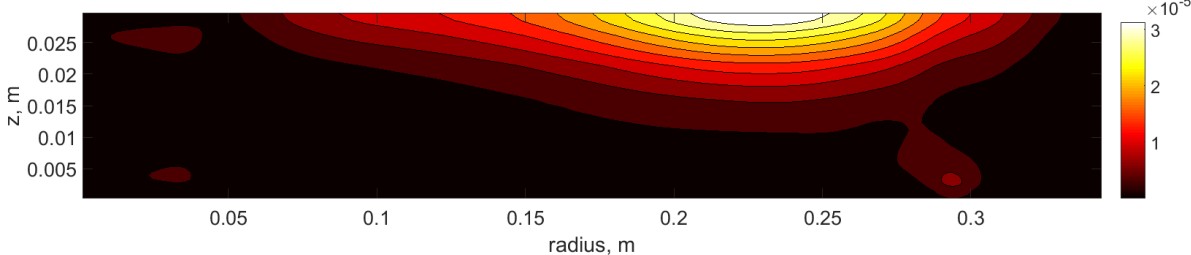

**Figure A3.** Vertical field of energy of radial velocity fluctuations in case with the polar heating. Numerical simulation. (Colour online)

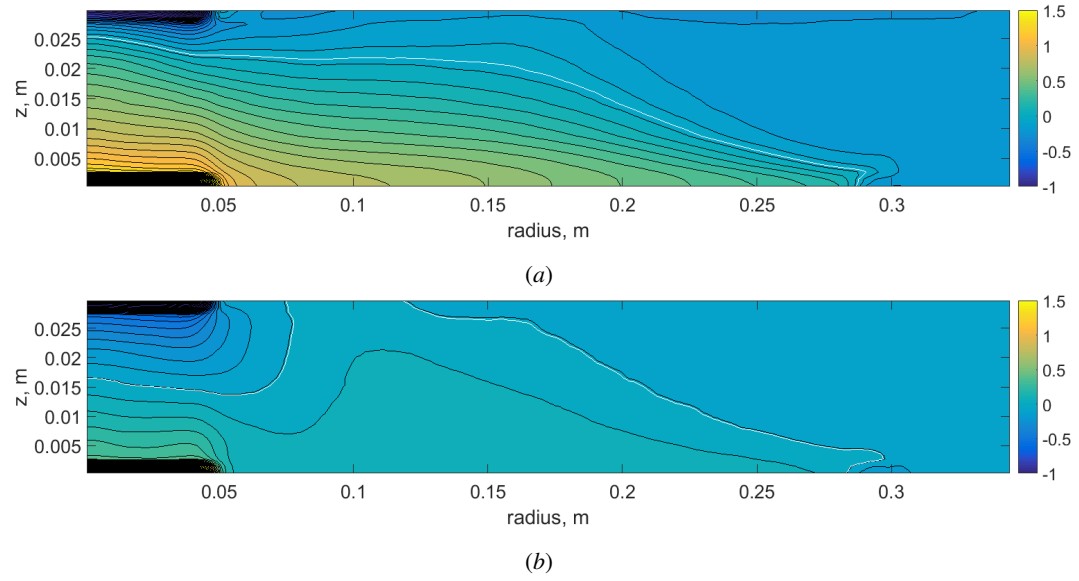

$(a)$

$(b)$

**Figure A4.** (a) – temperature difference $\Delta T_1 = T_{lcph} - T_{lcs}$, where $T_{lcph}$ – mean vertical temperature field with a large non-slip cooler at the top and large polar heater of the same power at the bottom, $T_{lcs}$ – mean vertical temperature field with a large slip cooler; (b) – temperature difference $\Delta T_2 = T_{lcph} - T_{nc}$, where $T_{nc}$ – mean vertical temperature field without localized cooling large. Numerical simulation. (Colour online)

*Acknowledgements.* We would like to thank the anonymous referee and Tim Woollings for their valuable comments that helped improve the manuscript. The study was done under the RSF project 22-61-00098.

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
