# Peer review of "The study of the impact of polar warming on global atmospheric circulation and mid-latitude baroclinic waves using a laboratory analog"

_EGUsphere, 2023_

## Author Comment (AC1)

We are grateful to the referees for their detailed comments that helped us to clarify important issues and to improve our manuscript. Below we provide answers to the referee's comments.

RC1: 'Comment on egusphere-2023-2797', Anonymous Referee #1, 19 Mar 2024

General comments:

The manuscript addresses a very interesting and timely problem, and investigates a laboratory model of polar amplification in a laboratory setting. The experimental apparatus, developed recently by the authors is a novel, and surprisingly realistic model of the atmospheric circulation of a hemisphere of Earth (in this case, the Southern hemisphere, as the tank is rotating in the clockwise direction). This setup with the applied thermal boundary condition is a significant improvement to the widely used baroclinis annulus settings, introduced by the groups of Fultz and Hide in the 1950s. To the best of my knowledge, this is the very first such differentially heated rotating setting that is able to capture qualitatively the three-cell convection of a hemisphere (baroclinic annulus models are typically restricted to one-cell sideways convective meridional flows). The two experiments, in which the Rossby waves and jets were traced using aluminum flakes are supported by a series of numerical simulations as well, which yielded a superb agreement with the observed surface patterns, and also revelaed the likely structure of the three- (plus one) cell dynamics and the strength of the zonal flow. The most important finding of the paper is surprising: the mean temperature field is reorganized in such a way in a polar warming scenario that the central domain and the larger part of the lower layer became warmer, whereas the upper layer and the "equatorial" part of the lower layer became colder. I would remark that this type of dynamics may be of relevance for the deeper understanding of the ongoing climatic processes: in the present-day global warming at the lower levels at the polar region a marked warming is observed in the lower layers, but at the tropics the warming indeed happens at higher levels of the troposphere/stratosphere, which is rather similar to what is observed here, and therefore, I believe that this finding is rather significant, as it clearly demonstrates that even in a fluid dynamic model like this one, where latent heat is not involved, this situation can develop. Therefore I believe that this is an important paper which should be published. The presentation is clear and concise, the paper is well written and the language is easy to follow.
Specific comments:

- The simulations applied a constant heat flux boundary condition at the free surface, whereas, in reality, I believe that a boundary condition with a heat flux given by $dT(x,y)/dt = 1/tau * (T0-T(x,y))$ would be more realistic, with $T(x,y)$ being the surface temperature field, $T0$ the room temperature and tau is the timescale determined by the material properties (e.g. specific heat) of the fluid. Certainly not for the present paper, but in the future it may be interesting to see whether such a surface forcing would give different results in the numerical simulations.

We agree that due to the non-uniform temperature distribution, heat flux determined from the temperature difference between the surface and the surrounding environment better approximates the laboratory system. In fact, we have already implemented such a scheme in the ongoing study and compared the results for different boundary conditions on the top surface (at the same heating power). We can conclude that there are some quantitative differences, but overall the results are very similar. Hopefully, we will present these results soon.

- The nice images in Fig.4 suggest that using video recordings of these patterns and PIV software such as VidPIV it would be possible to get measurment data for the mean zonal and meridional surface velocities in the experiment. Was such analysis conducted, or can it be done relatively easily? If yes, I think it would be nice to compare those from the simulated mean zonal and meridional fields, to see, e.g. whether the size of the cells are indeed such as seen in the numerical results.

We agree that realization of the PIV measurements and direct comparison of experimental and numerical data would be quite useful. We have a long time experience of using PIV to study optically transparent flows. During this study we found out for the rotating tank of this size it is not very easy. At first we used the aluminum flakes patterns for velocity field reconstruction, but in that case the spatial resolution is low, we can resolve the largest vortices and estimate the velocity of the mean flow. The characteristic values of velocity at the surface in experiment and numerical simulations are close. After that we realized a series of measurements using PIV particles using constant laser (time-resolved PIV) and 16 Hz camera but only in a sector of about 120 degrees (some special efforts are needed to illuminate the whole surface). On the base of this data we are going to compare characteristics of baroclinic waves (velocity of wave propagation, dominant wave number etc.) in experiment and numerical simulation.

- See my general comment above: I think it is very important that this experiments provides similar patterns than the actual atmosphere subject to polar warming. I believe that it would underline the significance of the paper if some discussion on this would be given in the Conclusions section.

We provided more stress on this issue in our conclusions.

Technical corrections/suggestions:

Figure 1b: In color it is nice, but in the printed (grayscale) version the blues and reds in the azimuthal flow field cannot be destinguished. I would suggest to replot this with similar light-to-dark scales that are used in pretty much all the other such diagrams of the paper.

At the final stage, if our manuscript would be accepted for publication, we replot all figures according to the journal requirements.

Line 93: The wording here is a bit misleading, when it says "The experimental model is a rectangular tank..." instead, I would suggest to reformulate the sentence like this: "The experimental model is a tank of a rectangular cross-section....", as the tank itself is not rectangular but cylindrical.

Corrected.

Line 200: I'm not sure about the terminology here. The text says: "leads to an additional circulation known as the Ekman pumping". I think Ekman pumping specifically refers to the extra downwelling flow created here, as a consequence of Ekman transport, but the circulation itself, I believe is not to be called Ekman pumping.

Yes, the Ekman pumping is a driver of this circulation. Corrected.

**RC2**: 'Comment on egusphere-2023-2797', Tim Woollings, 28 Mar 2024

This paper describes a set of lab experiments and accompanying numerical solutions as potential analogues to the problem of amplified Arctic warming. The experiments are novel and the results interesting. I believe the paper ultimately has promise, even if only as a proof-of-concept, but I do have some major concerns over the current framing of the work.

Major comments:

1. The main conclusion that a transformation of polar cell structure is a plausible explanation for Arctic amplification is not justified. This is based on the temperature structure with warming over the pole near the surface, but the forcing seems very different in the experiments (changes in upper level friction, cooling and strong descent) to the real context (changes in surface heat fluxes associated with sea ice loss, among several other mechanisms). If the authors want to make this argument, perhaps as one of several contributing mechanisms, it should be supported with a discussion or comparison to recent observed trends or future projections, to see if there are any similar changes in the polar cell in that context.

Presumably the numerical model is quite flexible, so one way to test this would be to compare with numerical experiments adding an additional heating at the polar surface, instead of changing the upper level cooling.

To address this comment, we performed numerical simulations with a large non-slip cooler at the top and a large central heater at the bottom of equal power and size. More details of this simulation are presented in Appendix. The structure of the polar cell for the case without cooler and for the case with compensated heater are different but the temperature increase in the lower layer is quite similar. Adding of the bottom heating and turning off the top cooler results in close temperature profiles, except for the central area above the heater. We can conclude that for the laboratory system the overall cooling in the central area plays crucial role for the lower layer temperature distribution.

2. I would question the significance of the changes in mode energies which are highlighted as another key result (9b). There appear to be some differences here but the contribution of internal variability should be quantified to test this. This could involve adding error bars to fig 9c, and/or similar figures, to show whether differences are statistically significant.

Despite the focus on the meridional cells in the text, the eddy results are potentially very interesting in that the changes are relatively small. An emerging consensus from climate modelling is that the storm track response to Arctic warming is weak (eg https://www.nature.com/articles/s41612-023-00562-5) and an alternative interpretation of the current results is that these very different experiments also support this conclusion.

We checked the convergence of the mean values of total energy of meridional velocity fluctuations and energies of separate modes. The variations of the mean value of total energy for the time interval larger than 6000 s is about 1%, so we can assume that Fig.9a is correct and should not change if we prolong our simulations. The dynamics of the baroclinic modes is more complex, there are some internal variations, slow and non-periodic, when one or another mode dominate and we do not achieve the convergence of the mean values of separate modes. In order to illustrate this we added 5% errorbars in Fig.9b. Please note there are also some

quantitative changes in Fig.9b, in new version we show energy for different modes averaged over interval from r=0.19m to r=0.23m (area of maximal baroclinic wave activity).

Minor comments:

1. The title should be adjusted to include the nature of the work, ie concerning laboratory analogues

The new title:

The study of the impact of polar warming on global atmospheric circulation and mid-latitude baroclinic waves using a laboratory analog.

2. The introduction should mention some of the literature on storm track responses to Arctic warming.

Added.

3. It should also note the central role of anthropogenic climate change in observed and modelled Arctic warming, as reported in many studies and IPCC reports etc.

We noted the role of anthropogenic influence.

4. Note also the work of Blackport and Screen casting doubt on claims of increased zonal flow meandering - eg DOI: 10.1126/sciadv.aay2880.

We describe on more details different views of the influence of the Arctic amplification on the mid-latitude zonal flow and waves.

5. The introduction motivates the study as attempting to resolve uncertainty around the transition between regular and irregular baroclinic waves in lab experiments. This doesn't seem particularly relevant and the paper doesn't really answer this anyway. I would have thought a better motivation is simply to test Arctic warming-like changes in a broader range of physical situations.

We agree with this comment and focus on the modeling of the Arctic warming scenario.

6. Fig 1 is confusing, partly due to the caption. Presumably the greens in panel a are speeds in the meridional plane, while the shading in b is the zonal wind? What are the lines in panel a, just indicative directions? It doesn't appear to be streamfunction.

We clarify the caption to the Fig.1.

7. Relatedly, 'azimuthal' is used repeatedly as well as meridional and zonal. It would be good to simplify and stick with one set of words - I suggest meridional and zonal.

For clarity we had replaced "azimuthal" by "zonal".

8. Fig 1 also shows relatively large vertical velocities, which is at odds with the real atmosphere in which there is a clear scale separation between horizontal and vertical velocities associated with the aspect ratio. This contributes to doubts over how realistic the polar cell changes are.

We agree that the results obtained in the laboratory cannot be directly applied to the real system. There are some limitations to experimental modeling, for example, we cannot realize the aspect ratio as in the real atmosphere. Our results give some hints on what to look for in full-scale simulations and observational data.

9. How long are the simulations compared to a typical baroclinic wave timescale in the system?

The typical timescale for the baroclinic wave in our simulations is about 5-7 rotation periods (T=17 s), which is about 100 s. The time of simulation (in a statistically steady-state) is at least 7600 s. Answering on the previous comment we noted that the time of simulation is sufficient for the calculation of the mean and fluctuating characteristics of the flow, but is not long enough to obtain mean characteristic of separate modes.

10. What is actually plotted in fig 5?

The lines are trajectories of fluid particles (streamlines) and the shading characterizes temperature distribution (white -- hot, black -- cold), the black circle in the middle -- the cooler.

We added description to the caption.

11. Around fig 6 - when introducing the different cases, it would be nice to discuss which of these are more or less realistic and in what ways.

We added some discussion of the role of the boundary condition on the cooler surface.

12. Lines 213-14: the largest differences actually seem to be in the polar cell, not the central one.

We changed the central cell to the laboratory polar cell for clarity.

13. Fig 8 is of limited use in telling the difference between simulations.

We agree that distributions for all cases are similar. We left one for the example of the vertical distribution of the energy of radial velocity fluctuations and its change after turning off the cooling.

14. The cooling seen at upper levels and larger radius (eg fig 13) is in striking contrast to observed and modelled Arctic warming. This is presumably a consequence of the fixed mean temperature and heating/cooling applied, so some discussion of this should be given. The real system is of course not in equlibrium due to the warming effect of greenhouse gases.

The laboratory model has its limitations and fixed boundary conditions for the heat flux can be the cause of the substantial temperature decrease in the upper layer in the periphery (low latitudes). Although there is a weak temperature decrease in low latitudes in the upper layer as a response for the modeled Arctic sea-ice loss in Ye et all.2024.

15. It's nice to see the temperature measurements from the physical experiment in fig 13c. These do show some discrepancy from the numerical model though, is there a hypothesis to explain this?

The temperature profiles in fig 13c show that numerical results for the large cooler are closer to the experiment with a small cooler. According to that we assume that the main source of discrepancy is underestimation of the cooling power in experiment. Unlike heating power, which we measure and control relatively easy, the measurement of total cooling power is not trivial. Since we focus mainly on qualitative description of the system we believe that it is not crucial to our analysis.

---

## Author Response (AR2)

We are glad that our manuscript is accepted for publication. Below we provide answers to the editor's comments.

1. The errorbars in figure 9b are very useful. Please could you just add the description of these to the caption (ie 95% confidence interval) so that it's in the paper as well as the response.

The necessary description is added.

2. Reading the abstract again I wondered if the authors wanted to highlight the importance of the polar cell more. Eg in lines 8-9 it might not be clear to many readers that it is the polar cell which is changing here as opposed to the eddies. Since this is the main result of the paper, I though it might be nice to state this explicitly here.

We agree with the comment and clarify the abstract.

We also checked our figures using the Coblis Color Blindness Simulator (https://www.color-blindness.com/coblis-color-blindness-simulator/) and revised the colour schemes accordingly.